# Fracture-Specific and Conventional Stem Designs in Reverse Shoulder Arthroplasty for Acute Proximal Humerus Fractures—A Retrospective, Observational Study

**DOI:** 10.3390/jcm10020175

**Published:** 2021-01-06

**Authors:** Jan-Philipp Imiolczyk, Philipp Moroder, Markus Scheibel

**Affiliations:** 1Department for Shoulder and Elbow Surgery, Center for Musculoskeletal Surgery, Charité-Universitaetsmedizin Berlin, 13353 Berlin, Germany; jan-philipp.imiolczyk@charite.de (J.-P.I.); philipp.moroder@charite.de (P.M.); 2Department of Shoulder and Elbow Surgery, Schulthess Clinic, 8008 Zurich, Switzerland

**Keywords:** reverse shoulder arthroplasty, proximal humerus, fractures, stem design, greater tuberosity, lesser tuberosity, fracture stems, lag sign, 155°, rehabilitation

## Abstract

Tuberosity healing and stem design can be outcome-dependent parameters in hemiarthroplasty for proximal humerus fractures (PHF). The relevance of fracture-specific stem design in reverse shoulder arthroplasty (RSA) is still a matter of debate. This retrospective study evaluates tuberosity healing and function for fracture specific stems (A) compared to conventional stems (B) in RSA for complex PHF in 26 patients (w = 21, mean age 73.5 years). Clinically, range of motion (ROM), Constant-Murley-Score (CS), Subjective Shoulder Value (SSV), and external rotation lag signs (ERLS) were evaluated. Healing of greater tuberosity (GT) and lesser tuberosity (LT), scapular notching, and loosening were examined radiologically. There were no statistical significant differences with regards to CS (A: 73 ± 11; B: 77 ± 9 points), SSV (A: 78% ± 11%; B: 84% ± 11%), external rotation (A: 18° ± 20°; B: 24° ± 19°), or internal rotation (A: 5.7 ± 2.2; B: 6.7 ± 2.8 CS-points) (*p* > 0.05). Mean forward flexion was superior for group A (*p* = 0.036). Consolidation of GT (82%) and LT (73%) was similar in both groups. Anatomical healing was slightly higher in group B (*p* > 0.05). Scapular notching was found in 27% (A) and 55% (B) (*p* > 0.05). RSA for PHF provides good to excellent clinical results. The quantitative and qualitative union rate for both cohorts was similar, indicating that fracture stems with open metaphyseal designs to allow for bone ingrowth do not improve tuberosity healing. ERLS correlates with a worse function in CS and ROM in all planes.

## 1. Introduction

Accounting for approximately 5% of all fractures, proximal humerus fractures (PHF) are one of the most common fractures shoulder surgeons must take care of. The incidence of PHF increases with age [1]. Most fractures only show minimal displacement and can be treated conservatively. More complicated and dislocated fractures though, often require surgical treatment [2,3]. Historically, hemiarthroplasty (HA) was considered the standard surgical option for non-reconstructable fractures due to poor bone quality and a high risk of non-union, dislocation, or osteonecrosis when treated with open reduction and internal fixation (ORIF) [4,5,6,7]. In the last few years however, reverse shoulder arthroplasty (RSA) has emerged as a reliable surgical alternative [8]. Patients treated with RSA for severely displaced three or four-part PHF showed encouraging results with a reduction of pain, good recovery in range of motion, and good functional outcome in midterm results [9,10,11,12,13]. While there is still controversy surrounding the correct surgical treatment of an acute PHF, evidence seems to support RSA over HA in most cases [14,15,16,17,18].

In HA for PHF, the healing of tuberosities is a strong indicator of success while a displacement of tuberosities usually results in malfunction [5,16]. Fracture specific stem design in HA results in greater tuberosity (GT) healing. This correlates with significant better external rotation (ER), forward flexion, and function [19,20].

Physiological GT alignment also appears to impact function in RSA [21] however, the outcome-dependency on tuberosity healing seems to be lower [16,22,23,24].

In general, tuberosity healing in RSA can be dependent on many different factors like humeral inclination, lateralisation, torsion of the humeral component, retroversion of the glenoid component, or the tuberosity fixation technique [21,25,26,27,28,29,30].

Limited studies are already available, however more extensive research on the influence of stem design on tuberosity healing, clinical function, and patient satisfaction is necessary [31,32].

Therefore, the aim of this study was to compare a fracture specific stem to a conventional stem in primary RSA for the treatment of acute, complex PHF with regards to clinical and radiological results in patients with the same glenoid reconstruction, same tuberosity refixation technique, and same restrictive rehabilitation protocol.

## 2. Patients and Methods

### 2.1. Study Design

In a retrospective cohort study, 35 consecutive patients with complex, non-reconstructable acute PHF were treated with RSA from December 2009 until May 2018.

Inclusion criteria were patients with non-reconstructable, displaced PHF treated by a specialised shoulder surgeon within a maximum of 14 days after trauma that were not treated by ORIF or HA. Exclusion criteria were patients with PHF that had been treated non-operatively, have had any shoulder operations previous to the fracture, or had displayed signs of axillary nerve damage.

All patients were treated in our hospital by the two senior surgeons (M.S. and P.M.). Patients received a Grammont type of prosthesis (155° of humeral inclination) with either a fracture specific stem design with open metaphysis allowing for bone ingrowth (Group A) (Aequalis Reversed FX, Wright Medical Group Inc., Memphis, TN, USA) or the conventional stem (Group B) (Aequalis Reversed II, Wright Medical Group Inc., Memphis, TN, USA) (both shown in Figure 1 in true a/p radiographs).

The fracture specific stem (Figure 1b) is a monobloc stem with Hydroxyapatite coating (or bone ingrowth encourager) and a metaphyseal window for bony augmentation to encourage fracture and especially tuberosity healing. Furthermore, the slim diaphyseal and metaphyseal design allows an easier humeral implantation. The conventional stem (Figure 1e) consists of two components, a cobalt-chrome diaphyseal stem and a metaphysis with anti-rotation design.

For both stems, polyethylene cups in different sizes are available. All stems were cemented.

### 2.2. Surgical Technique and Rehabilitation Protocol

All surgeries were performed in beach-chair position using a deltopectoral approach. A biceps tenotomy was performed in all cases. After the removal of the humeral head, two #5 FiberWire^®^ sutures (Arthrex, Naples, FL, USA) were placed through infraspinatus and teres minor close to the greater tuberosity and two #5 FiberWire^®^ sutures (Arthrex, Naples, FL, USA) were placed through the subscapularis at the bone-tendon junction at the lesser tuberosity. In total, four cerclages were used as holding sutures in the osseus-tendinous junction to hold the lesser and greater tuberosity. This allowed an anatomical reduction of tuberosities, even in case of comminution. The supraspinatus was torn or partially torn in nine patients and excised in all others. There were no full-thickness tears of the anterior or posterior rotator cuff present at surgery. After the exposure of the glenoid, a standard 36 mm glenosphere was implanted in 14 cases and an eccentric glenospheres in 12 cases. After preparation of the humerus and implanting the cemented stem, the fractured tuberosities were reattached, tensioning the sutures in a belt-like technique around the metaphyseal neck of the prosthesis under image-intensifier control. An additional medial FiberWire^®^ cerclage (Arthrex, Naples, FL, USA) was used to prevent the butterfly effect (Figure 2). All patients received bone grafting from the humeral head to encourage bone healing of the greater and lesser tuberosity. When thin, egg-shell like or comminuted tuberosities were present, cancellous bone was placed underneath. For the fracture-specific stem, additional bone grafting within the metaphyseal fenestration was performed.

After surgery, the operated shoulder was strictly immobilised in a neutral position brace for 14 days. Active wrist and elbow movement was allowed during this time. Passive mobilisation of the shoulder started at week three and active mobilisation six weeks after implantation. All patients followed the exact same rehabilitation protocol.

### 2.3. Clinical and Radiologic Evaluation

All patients were clinically and radiologically evaluated at each follow-up examination after six weeks, three months, and then every year when possible.

Clinically, Constant-and-Murley Score (CS) [33], Subjective Shoulder Value (SSV) [34], and range of motion (ROM) of each patient were documented. Abduction, forward flexion, and external rotation were documented in degrees, while internal rotation was rated using the CS scale. We evaluated pain using a visual analogue scale (VAS) (0–15 points) and abduction strength was measured using IsoBex (Herkules Kunststoff AG, Oberburg, Schweiz) in 90° of abduction in the plane of the scapula. In addition, the presence of an external rotation lag sign (ERLS) was documented, when passive ER in 0° abduction was greater than active ER. ERLS was documented when the arm dropped back from passive ER to 0° or less than 0° (ERLS++) and when active ER was still possible, passive ER was greater by at least 10° (ERLS+).

All patients received standard radiographs (true a/p, Y-view and axillary view) to evaluate tuberosity healing and scapular notching according to Sirveaux [35]. Patients with unclear tuberosity healing received additional radiographs in true a/p in external and internal rotation for evaluation. Tuberosity healing was divided into anatomical healing, malunion and non-union, or resorption (Figure 3). Furthermore, osteophytes, loosening, radiolucency, condensation lines, thinning of the lateral cortex of the proximal humerus, and formation of scapular bone spurs were included in radiologic evaluation. The radiographs were evaluated by two independent observers (M.S. and J.-P.I.).

### 2.4. Bias

This study was not blinded and observer bias due to subjectivity is possible. In CS sections, activities and sport were adapted to age and referenced to preoperatively and therefore the maximum amount of points could be achieved. For SSV, patients have usually used the other arm as reference of 100%, which does not equivalate the strength and function of an athletes’ 100%. There is a selection bias present amongst our study as patients were treated by one of two surgeons who were not at the beginning of the learning curve implanting RSA. Patients were treated using an established tuberosity fixation and showed compliance agreeing to a restrictive rehabilitation protocol for six weeks.

### 2.5. Statistical Analysis

For comparability analysis, binomial values like gender, side, and presence of comorbidities were evaluated using the Fisher’s exact test. For continuous parameters like age, fracture morphology, and follow-up time an unpaired t-test was used. Due to the relatively small sample size, non-parametric tests (Mann-Whitney-U) for independent samples were performed with SPSS 27 (IBM, Armonk, NY, USA). Quantitative variables were described by means, standard deviations, minimums, and maximums. We have investigated differences in clinical (CS, SSV, and ROM) and radiographic outcome with regards to both stems. Prognostic factors for functional outcome in CS-subgroups, ROM, and SSV were tested against tuberosity healing/consolidation and for ERLS. Significance level was set to 0.05.

## 3. Results

Overall, we treated 35 patients with a mean age of 74.5 (range: 59–87) years, eight men and 27 women (77%) were included. A total of 16 patients presented a severe III- and IV-part PHF and in 10 additional cases a head-split was present. Six patients suffered a dislocation fracture and three patients with osteoporosis a displaced II-part subcapital PHF was treated with a RSA. Five of the above mentioned had an additional antero-inferior glenoid fracture.

Seven patients had to be excluded from the final evaluation due to death (*n* = 5) or limited general health status prohibiting an outpatient clinic visit (*n* = 2). Two patients had to be excluded because of revision surgery. This left 26 patients for a final follow-up, four of which had insufficient radiographic data. A study flow-chart is displayed in Figure 4. The demographics of these patients and fracture morphologies for both groups are displayed in Table 1.

### 3.1. Comparison of Clinical Results

Both groups showed very good results for RSA after acute PHF (Table 2, Figure 5 and Figure 6).

At A final follow-up, an average CS of 72.6 ± 11.0 points and SSV of 78.3% ± 10.9% was achieved in group A. In group B, a slightly better average CS of 76.9 ± 8.8 points and SSV of 84.3% ± 10.7% was achieved at the final follow-up. Functional results are summarised in Table 3.

Only active forward flexion presented significantly better results for patients treated with the conventional stem. In Figure 5, results for CS and active forward flexion are displayed with regards to follow-up time.

In group A, two patients (17%) displayed an ERLS+ and two patients (17%) an ERLS++ was documented. In group B, four patients (29%) displayed an ERLS+ and in two more patients (14%), an ERLS++ was observed (*p* = 0.14).

### 3.2. Comparison of Radiologic Results

In both groups, consolidation of GT was achieved in 82% and of LT in 73%.

Anatomical healing, however, was higher for GT (group A: 55%; group B: 64%) and LT (group A: 64%; group B: 73%) for patients treated with the conventional stem (shown in Table 4).

Scapular notching was documented in three cases (27%) in group A, two patients showed grade 1 notching and in one patient grade 3 was visible. In group B, scapular notching was documented in six cases (55%), all grade 1 (*p* = 0.20). As both stems are based on a 155° neck shaft angle, notching occurred more often in patients with non-eccentric glenospheres (*p* = 0.045). Furthermore, eccentric glenosphere show statistically greater ER (29° vs. 15°; *p* = 0.033) but also lower abduction strength (4.3 vs. 3.4 kg; *p* = 0.047).

Thinning or resorption of the lateral cortex of the proximal humerus was observed only in group B (*n* = 2). First signs were visible in both patients at six years follow-up. However, those two patients showed excellent function (one of them shown in Figure 6) with significantly greater forward flexion (*p* = 0.03), abduction (*p* = 0.02), and SSV (*p* = 0.02) but significantly more pain (*p* = 0.01).

Scapular bone spurs were found in five patients for each group. Scapular bone spurs were associated with a greater abduction strength (*p* = 0.06), however not significantly.

In both groups, we did not find any radiolucency, loosening, or condensation lines.

### 3.3. Clinical Impact of Tuberosity Healing and ERLS

For healing of the GT, only anatomical healing has shown a significant improvement for SSV (86 vs. 75%; *p* = 0.028), external rotation (27° vs. 9°; *p* = 0.022), and less pain (CS points 14.8 vs. 13.7; *p* = 0.033).

Anatomical LT healing did also correlate significantly with greater ER (26° vs. 6°; *p* = 0.008). A non-union of LT correlated significantly with lower forward flexion (141° vs. 158°; *p* = 0.019) and abduction (141° vs. 153°; *p* = 0.029).

Our data suggests that an ERLS, with a drop to 0° or less (++) or an ERLS with passive ER greater than active ER by a minimum of 10 degrees (+), exhibits prognostic value and correlates with worse function and outcome in RSA (Table 4).

### 3.4. Complication

We examined the electronic medical records and scanned for adverse events. We found complications in two patients. One glenoid component fracture that was treated with a revision of the glenoid component (Group A) and one dislocation during physiotherapy treated operatively with an inlay revision from +9 to +12 (Group B). Both patients have been excluded from this analysis, resulting in a revision rate of 6.7% (A) and 5.0% (B).

## 4. Discussion

Elderly patients with possible cuff pathologies, along with patients suffering from complex fractures seem to benefit from RSA the most [36,37,38].

Although tuberosity integration in RSA is not as important for function as with HA, when a current metanalysis has compared fracture-specific stem design for HA and RSA mixed together, it has shown a favourable outcome for fracture stems regardless of different rehabilitation protocols or tuberosity refixation techniques [32].

Our study shows that a conventional stem could achieve similar clinical results for complex PHF like a contemporary, bone-ingrowth, fracture-specific stem. Apart from favourable forward flexion with the conventional stem, there has been no significant differences in clinical or radiographic outcome parameters. GT union results in higher subjective satisfaction and less pain with greater range of motion in external rotation. LT non-union correlates with lower forward elevation and abduction of the arm.

Our data supports those findings of Grubhofer et al. [39] and Boileau et al. [40] that patients with GT healing improve in external rotation, forward flexion, and subjective function.

This higher rate of anatomical tuberosity healing in group B could account for the better results on average in CS, ROM, and SSV because according to theoretical models, GT healing does improve stability through deltoid wrapping [41]. Tuberosity healing has been described in the literature with similar GT healing of 65–84% [21,39,40,42,43] comparable to a rate of 82% for both group A and B.

Both analysed stems are Grammont designs with the same inclination (155°) of the humeral component. The main difference of stems is the open design of the metaphysis in the fracture stem as well as the flat lateral flange and the medial calcar hole for the suture passage. These design features, in theory, should allow better fixation and bony ingrowth of the tuberosities. However, the quantitative and qualitative union rate for both cohorts was similar, indicating that fracture-specific stems do not improve tuberosity healing despite their theoretical prosthetic advantages. It could be argued that our tuberosity refixation supports healing also around the broad metaphyseal neck of the conventional prosthesis.

Therefore a strong tuberosity refixation technique with a restrictive rehabilitation protocol seems to be mandatory.

Our results indicate that LT consolidation has a stronger impact on subjective and objective outcome than imagined. LT healing strongly correlates with greater external rotation, non-union of LT accounts for lower forward flexion and abduction (*p* < 0.05). A possible explanation is that LT healing was often achieved in combination with GT healing and therefore despite there being a stronger correlation, it is not necessarily due to causation that LT healing improves ER.

Our data shows that eccentric glenospheres correlate with fewer scapular notching, greater external rotation but also less abduction strength (*p* < 0.05). Those patients with total GT resorption and even further lateral thinning perform interestingly significantly better in SSV, arm elevation, but also state a slightly higher value on the pain scale (*p* < 0.05). Scapular bone spurs correlate with greater strength in abduction (*p* = 0.06). Those radiographic phenomena might be some kind of indicator for biomechanical changes due to deltoid remodelling.

The clinical ERLS proposes high prognostic value in our study. Patients without ERLS did show significant greater CS, ROM in all planes and strength (*p* < 0.05). This trend towards higher SSV when no ERLS is present (*p* = 0.06) is based on the fact that ER is essential for daily activities (combing hair, putting on clothes, washing, etc.) [44].

As our patients perform better on average (CS: 74 points) as previously described in the literature (mean CS: 60–64 points) with similar refixation techniques [21,39,40,42,43] this might be due to our selection bias. Patients with revisions were excluded, only specialised shoulder surgeons with extensive training and experience performed surgery. Some patients were lost to follow-up either due to limited general health status or due to dementia, questioning the compliance whether a strict immobilisation in a brace for 14 days in 0° external rotation is necessary or even realistic.

However, compared to different studies, rehabilitation protocols range from pain-adapted active movements beginning Day 1 [45], two days of immobilisation [9], internal rotation slings for four to five [46] or up to six [39,43] weeks, or abduction pillows for six weeks [21]. As all patients in our cohort performed relatively better, it could be argued that our rehabilitation with a strict immobilisation in 0° ER, reducing stress on tuberosities, provides superior results. A 155° inclination and overall medialisation as well as our strict neutral positioned immobilisation are factors reducing the stress on tuberosities [47]. This might account for better ingrowth of elderly porous bone to the prosthesis, therefore higher stability and better function. The goal is to strictly immobilise long enough to allow for enough stabilisation but not too long to prevent stiffness and enable deltoid remodelling.

This study has several limitations. One of which is the very small cohort number of both groups with a relatively high rate of lost to follow-up (26%). This is due to the old and morbid patient cohort, which is also typical for other cohort studies of RSA for PHF with a dropout of about 29–34% [21,39,40].

Secondly, only two-dimensional radiographic evaluation for tuberosity healing was obtained. Additional radiographs were taken for better evaluation in some cases. Only a three-dimensional computer tomography could provide a more accurate evaluation of tuberosity integration with the expense of higher radiation exposure.

The difference in time of follow-up is another limitation that accounts for higher rates of scapular notching and resorption of the lateral cortex in group B, as those progress over time. Moreover, active forward flexion is significantly better with the conventional stem. However, as indicated in Figure 5, active forward flexion may not have improved enough in the first year after operation and therefore further follow-up is needed to underline this finding.

A strength of our study is, that we have used the same glenoid reconstruction, same tuberosity refixation technique, and same restrictive rehabilitation protocol for all patients in both groups in order to investigate the effect of different stem designs.

Furthermore, studies with larger patient cohorts are needed to investigate prognostic effects for this elderly patient cohort for acute PHF with regards to stem designs, rehabilitation protocol, cuff status, and tuberosity healing to provide more reliable clinical take-home messages.

## 5. Conclusions

Both groups showed very good clinical results for the treatment of complex, acute proximal humerus fractures. Although only active forward flexion was better with the conventional stem, this could be due to follow-up time bias. The quantitative and qualitative union rate for both cohorts was similar, indicating that fracture stems with open metaphyseal designs to allow for bone ingrowth do not improve tuberosity healing.

## Figures and Tables

**Figure 1 jcm-10-00175-f001:**
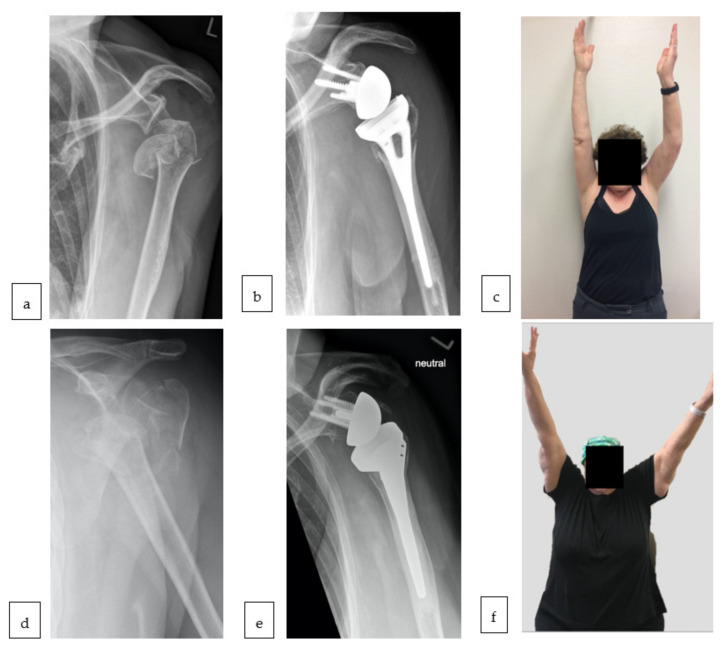
The figure show a 72-year-old female patient with a head split proximal humerus fractures (PHF) (**a**) treated with a fracture specific stem (**b**) four years after implantation (**c**). The figure shows a 70-year-old female patient with a four-part fracture (**d**) treated with conventional stem (**e**) five years after implantation (**f**).

**Figure 2 jcm-10-00175-f002:**
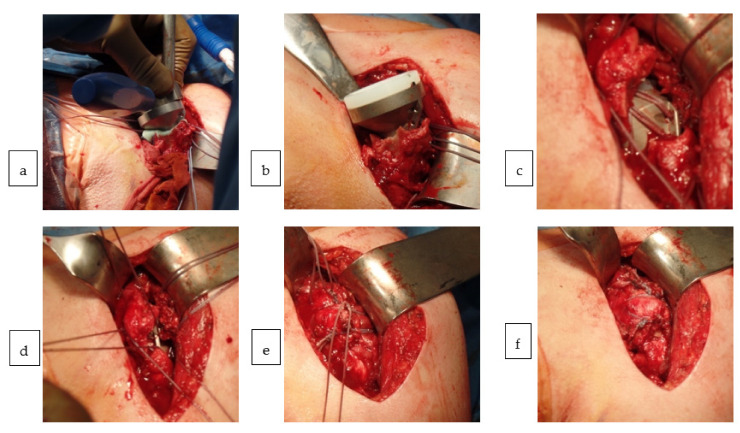
Tuberosity repair with the conventional stem: After fracture exposure and placement of four sutures, two through the anterior and two through the posterior cuff, cementation was followed with stem implantation (**a**). Removal of excessive cement allowing bony tuberosity integration around the metaphysis was then followed by inlay positioning (**b**). Tensioning the anterior and posterior sutures through the fins (only for the conventional stem) (**c**) before aligning both tuberosities around the metaphysis of the stem (**d**). Closing the anterior and posterior sutures (**e**) and preventing the butterfly effect through the use of an additional, medial cerclage (**f**).

**Figure 3 jcm-10-00175-f003:**
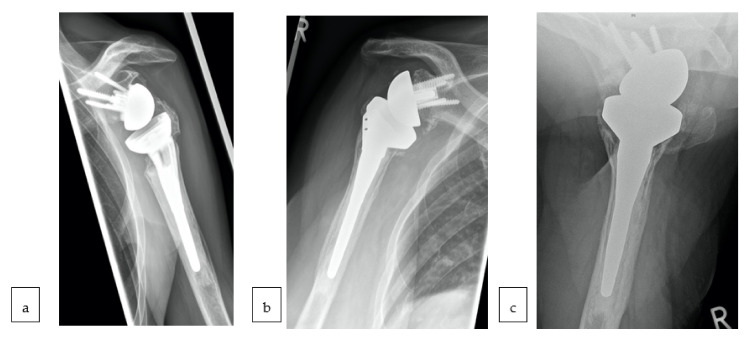
Tuberosity evaluation: A 76-year-old female patient treated with the fracture specific stem shows malunion of greater tuberosity (GT) and resorption of lesser tuberosity (LT) (**a**). A 68-year-old female patient with standard stem shows anatomical healing of LT and malunion of GT in true a/p (**b**) and axial (**c**) radiographs.

**Figure 4 jcm-10-00175-f004:**
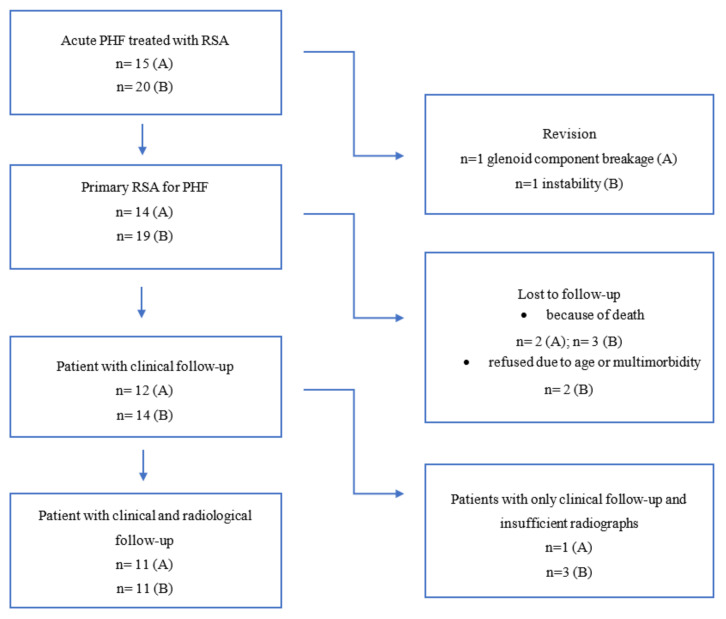
Study flow-chart with regards to patients treated with the fracture-specific stem (A) and the conventional stem design (B). (RSA—reverse shoulder arthroplasty, PHF—proximal humerus fracture).

**Figure 5 jcm-10-00175-f005:**
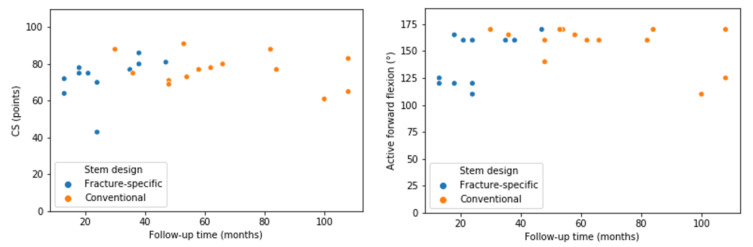
Scatter plots for CS and active forward flexion for both stems with regards to follow-up time. Function for CS and forward flexion are lower in those patients with early follow-up. This statistical significance might be biased due to further improvement in function in the second year after operation.

**Figure 6 jcm-10-00175-f006:**
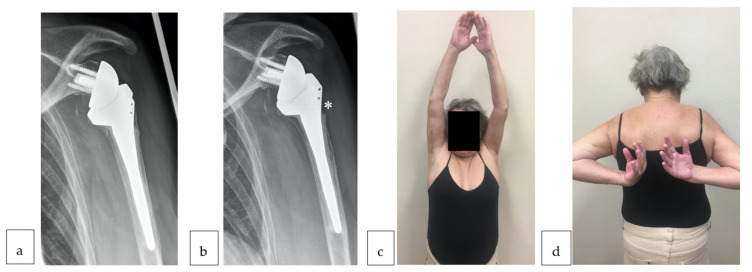
A 70-year-old female patient presents at a six-year follow-up GT resorption in true a/*p* radiograph (**a**). Three years later, further resorption of the lateral cortex of the proximal humerus (*) has occurred (**b**). She was not entirely pain-free (VAS 2/15), however, presented excellent clinical function in abduction (**c**) and internal rotation (**d**). Better results in forward flexion, abduction, and SSV but also more pain is associated with this cortex resorption (*p* < 0.05).

**Table 1 jcm-10-00175-t001:** Patient demographics and fracture morphologies with regards to all patients with clinical follow-up.

Variables	Fracture-Specific Stem (A)	Conventional Stem (B)	Total	*p*-Value
	*n* = 12	*n* = 14	*n* = 26	
Gender (*n*)	-	-	-	0.63
Men	2	3	5	
Women	10	11	21	
Age at surgery (years)	75.5	71.7	73.5	0.15
	(71–82)	(59–87)	(69–87)	
Clinical follow up (months)	26	67	48	< 0.001
	(13–47)	(30–108)	(13–108)	
Surgical side (*n*)	-	-		1
Right	4	6	10	
Left	8	8	16	
Dominant side (*n*)	-			0.7
Dominant	5	5	10	
Not dominant	7	9	16	
Comorbidities (*n*)	-	-		0.22
None	2	6	8	
1–2 comorbidities	2	2	4	
3–4 comorbidities	2	5	7	
5 or more comorbidities	6	1	7	
Fracture morphology	2× II-Part PHF	1× II-Part PHF	3× II-Part PHF	0.5
	3× III-Part PHF	3× III-Part PHF	6× III-Part PHF	
	5× IV-Part PHF	6× IV-Part PHF	11× IV-Part PHF	
	2× dislocation IV-Part PHF	4× dislocation IV-Part PHF	6× dislocation IV-Part PHF	
Additional head-split component	5	3	8	0.29
Additional glenoid fracture	1	1	2	0.92

A comparability analysis for both cohorts showed only significant differences for follow-up time. Fracture morphologies are classified according to Neer [3]. PHF—proximal humerus fracture.

**Table 2 jcm-10-00175-t002:** Comparison of clinical results at the final follow-up for both groups.

Variables	Fracture Stem (A) *n* = 12		Conventional Stem (B) *n* = 14		
	Average	Stand. Dev.	Average	Stand. Dev.	*p*-Value
CS (points)	72.6	11.0	76.9	8.8	0.21
SSV (%)	78.3	10.9	84.3	10.7	0.13
VAS	14.2	1.8	14.4	1.0	0.50
ADL	17.7	2.8	18.6	1.4	0.22
ABD (°)	142	24	155	18	0.12
FLEX (°)	144	23	157	19	0.036 *
ER in 0° ABD (°)	18	20	24	19	0.21
IR (points) **	5.7	2.2	6.7	2.8	0.09
Strength of ABD (kg)	3.9	1.1	3.9	2.1	0.49
ABD glenohumeral (°)	76.3	9.3	81.4	8.9	0.05

Stand. Dev.—Standard Deviation, CS—Constant Score, SSV—Subjective shoulder value, ABD—abduction, FLEX—forward flexion, ER—external rotation, IR—internal rotation, VAS—visual analog scale, ADL—activities of daily living, * statistically significant (*p* < 0.05), ** active IR scored with points: 0 = lateral thigh, 2 = buttocks, 4 = sacrum, 6 = L3, 8 = T12, 10 = T7–8.

**Table 3 jcm-10-00175-t003:** Descriptive table with quality of GT and LT healing with regards to stem.

Variables	GT		LT	
	A	B	A	B
Anatomic healing (*n*)	6	7	7	8
Malunion (*n*)	3	2	1	-
Non-union/resorption (*n*)	2	2	3	3

Tuberosity healing was only evaluated for those 22 patients with sufficient radiologic data (A: *n* = 11; B: *n* = 11). GT—greater tuberosity, LT—lesser tuberosity, A—fracture-specific stem; B—conventional stem.

**Table 4 jcm-10-00175-t004:** Comparison of subjective satisfaction and function with regards to patients that show a positive external rotation lag signs (ERLS).

Variables	No ERLS *n* = 16		ERLS +/++ *n* = 10		
	Average	Stand. Dev.	Average	Stand. Dev.	*p*-Value
CS (points)	78.3	7.2	69.4	11.5	0.021 *
SSV (%)	84.4	8.5	77.0	13.4	0.06
VAS	14.4	1.3	14.2	1.6	0.41
ADL	18.8	1.2	17.2	3.0	0.07
ABD (°)	158	10	133	27	0.007 *
FLEX (°)	159	15	138	24	0.014 *
ER in 0° ABD (°)	33	16	4	6	<0.001 *
IR (points) **	7.0	1.9	5.0	3.0	0.041 *
Strength of ABD (kg)	4.1	1.8	3.6	1.5	0.43

Stand. Dev.—Standard Deviation, CS—Constant Score, SSV—Subjective shoulder value, ABD—abduction, FLEX—forward flexion, ER—external rotation, IR—internal rotation, VAS—visual analog scale, ADL—activities of daily living. * statistically significant (*p* < 0.05). ** active IR scored with points: 0 = lateral thigh, 2 = buttocks, 4 = sacrum, 6 = L3, 8 = T12, 10 = T7–8.

## Data Availability

The data presented in this study is available on request from the corresponding author. The data is not publicly available due to privacy and ethical restrictions.

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
