# Peer review of "Fracture-Specific and Conventional Stem Designs in Reverse Shoulder Arthroplasty for Acute Proximal Humerus Fractures—A Retrospective, Observational Study"

_jcm, 2021, doi:10.3390/jcm10020175_

Round 1
Reviewer 1 Report
Manuscript title: Is a fracture-specific stem design of benefit in reverse shoulder arthroplasty for acute proximal humerus fractures? An observational study
I have some concerns that must be addressed.
Title: I suggest considering a change in the title structure to avoid it being a question and the addition of the word “retrospective”
Introduction: In this section, I suggest adding more information about some concerns like design features, i.e. humeral inclination, neutral versus lateralized versus inferiorized glenosphere, and their possible role in clinical outcome and greater tuberosity healing.
Line 50: “The influence of stem design and tuberosity healing in RSA have not been finally investigated”. the impact of stem design and tuberosity healing on function after RSA has received limited studies that you can find with a more extensive research
Materials and methods: I suggest dividing this section into numbered paragraphs, for example, “Study design”, “Surgical procedure and postoperative care”, “Statistical analysis” etc.
A more systematic approach should be used. Here are some of the important issues that have not been explored in the text. Have you followed the STROBE recommendation? where did you retrieve data? Did patients sign informed consent for the scientific use of their data? Did you apply any inclusion and exclusion criteria? During what period were the patients treated? Which patients had a glenoid fracture? More information on the surgical technique should be added. When did you review patients? and what did you evaluate during each consultation? At what follow-up did the patient receive the x-rays? and when did you make any additional radiographs? did you research for complications? Did you use Neer classification for preoperative radiographs? add the reference. How were the fracture patterns distributed in the two groups? how did you evaluate the reduction of the greater tuberosity in the postoperative radiography? and in subsequent radiographs? Did you evaluate strength? How did you score IR? Did you use VAS? Also, some data should be moved to the "results" section, for example, lines 58 – 66, tables 1 and 2.
In the results section, it would be useful to insert all data of the patient selection process and add them to the study flow chart diagram. You should also specify for each group (when possible with range), also in tables: mean age, the number of women and men, mean follow-up, mean delay before surgery, mean ASA score if possible, preoperative classification of the fracture. In general, all data should be reported more clearly, also editing the current paragraphs. For example, it is not clear why lines 208-209 are inserted in the radiological results paragraph.
I recommend improving the discussion section with a more organic exposition, offering a cautious overall interpretation of results, considering objectives, limitations, multiplicity of analyzes and other relevant evidence, placing them in context of previous findings, discussing the generalizability of the study results, and explaining what they mean for future research.
Moreover, the authors should add Author Contributions and funding
The paper must be checked by an English native speaker.
Author Response
Response to reviewers:
We thank the reviewer very much for his criticism. Furthermore, we are thankful for the detailed comments and inputs provided to improve our manuscript. We have addressed all comments and suggestions. We have implemented structural changes according to the STROBE recommendation and hope that you are satisfied with our changes.
Please find below our answers to the comments and attached also the revised manuscript.
Comments and Suggestions for Authors:
I have some concerns that must be addressed.
Title:
I suggest considering a change in the title structure to avoid it being a question and the addition of the word “retrospective”
Response: We agree with this correction.
Changes made: We have changed the title to include the addition of the word “retrospective” and avoid it being a question:
Fracture-specific and conventional stem designs in reverse shoulder arthroplasty for acute proximal humerus fractures– A retrospective, observational study
Introduction:
In this section, I suggest adding more information about some concerns like design features, i.e. humeral inclination, neutral versus lateralized versus interiorized glenosphere, and their possible role in clinical outcome and greater tuberosity healing.
Response: We agree that there are more design features that can improve tuberosity healing and/or function. We have held our introduction brief because we solely focused on the fracture specific stem design and its advantages. Nevertheless, other prognostic design features should be addressed in the introduction.
Changes made: "In general, tuberosity healing in RSA can be dependent on many different factors like humeral inclination, lateralization, torsion of the humeral component, retroversion of the glenoid component or tuberosity fixation technique (1-7).” (page 2 line 61-63)
Line 50: “The influence of stem design and tuberosity healing in RSA have not been finally investigated”. the impact of stem design and tuberosity healing on function after RSA has received limited studies that you can find with a more extensive research
Response: Thank you very much for this important point. The following changes have been made.
Changes made: “Limited studies are already available, however, more extensive research on the influence of stem design on tuberosity healing, clinical function and patient satisfaction is necessary. (8-9)” (page 2 line 61-63)
Materials and methods:
I suggest dividing this section into numbered paragraphs, for example, “Study design”, “Surgical procedure and postoperative care”, “Statistical analysis” etc.
Response: We have divided this paragraphs into: “Study design”, “Surgical technique and rehabilitation protocol”, “Clinical and radiologic evaluation”, “Bias”, “Statistical analysis”
A more systematic approach should be used. Here are some of the important issues that have not been explored in the text. Have you followed the STROBE recommendation? where did you retrieve data? Did patients sign informed consent for the scientific use of their data? Did you apply any inclusion and exclusion criteria? During what period were the patients treated? Which patients had a glenoid fracture? More information on the surgical technique should be added. When did you review patients? and what did you evaluate during each consultation? At what follow-up did the patient receive the x-rays? and when did you make any additional radiographs? did you research for complications? Did you use Neer classification for preoperative radiographs? add the reference. How were the fracture patterns distributed in the two groups? how did you evaluate the reduction of the greater tuberosity in the postoperative radiography? and in subsequent radiographs? Did you evaluate strength? How did you score IR? Did you use VAS? Also, some data should be moved to the "results" section, for example, lines 58 – 66, tables 1 and 2.
Response: Thank you very much for this input. We have rearranged the “material and methods” section according to the STROBE recommendation. We have retrieved all the data from our medical and surgical records in our clinic. All patients gave written consent, time frame (line 74) and inclusion and exclusion criteria were added to the manuscript. We have reviewed all patients at each follow-up examination clinically and radiologically. This happened after six weeks, three months, six months and then every year. This is why we were able to determine the point in time when resorption of the lateral cortex has occurred. However, some patients did only show up for follow-up examination every two years and some had lost interest after some follow-up appointments, so that we do not have enough data to compare the results for 1 year postoperative vs 2 years and so on.. Therefore we have only evaluated the last clinical examination of all patients available. We have taken additional radiographs for all patients where it was not clearly visible in true a/p and axial xrays. We have then performed additional ones in internal and external rotation however this was not standard procedure and happened in agreement with the patient. We have researched for complications in our medical records and our stationary files. Moreover, all patients treated were present for follow-up examination. Therefore we were able to state our complication in the manuscript under 3.4 (page 10 line 382ff). We have moved the data and table you have mentioned above to the result section. We have added a “Bias” paragraph according to STROBE recommendation (page 5 line 246ff) and also added some aspects for clinical and radiologic evaluation (page 4, lines 189ff) and for surgical technique (line 142ff, line 152ff).
Changes made:
page 2 lines 76-80;
“Inclusion criteria were patients with non-reconstructable, displaced PHF treated by a specialized shoulder surgeon within a maximum of 14 days after trauma that were not treated by ORIF or hemiarthroplasty. Exclusion criteria were patients with PHF that had been treated non-operatively, have not had any shoulder operations previous to the fracture or have displayed signs of axillary nerve damage.”
page 4 lines 189ff;
“All patients were clinically and radiologically evaluated at each follow-up examination after six weeks, three months and then every year when possible. Clinically, Constant-and-Murley Score (CS) [25], Subjective Shoulder Value (SSV) [26] and range of motion (ROM) of each patient were documented. Abduction, forward flexion and external rotation were documented in degrees, while internal rotation was rated using the CS scale. We have evaluated pain using a visual analogue scale (VAS)(0-15 points) and abduction strength was measured using a IsoBex (Herkules Kunststoff AG, Oberburg, Schweiz) in 90° of abduction in the plane of the scapula.”
In the results section, it would be useful to insert all data of the patient selection process and add them to the study flow chart diagram. You should also specify for each group (when possible with range), also in tables: mean age, the number of women and men, mean follow-up, mean delay before surgery, mean ASA score if possible, preoperative classification of the fracture. In general, all data should be reported more clearly, also editing the current paragraphs. For example, it is not clear why lines 208-209 are inserted in the radiological results paragraph.
Response: We agree with your suggestion and excuse the intransparent presentation of data. In the text we have mentioned all 35 patients with regards to gender, age and fracture morphology. We have added a study flow chart as Figure 4 on page 8. In addition we have specified in Table 1 preoperative data, including preoperative fracture morphology according to Neer with regards to both groups. We cannot however examine from our medical records if operation was one or two days after trauma. Unfortunately, we do not have mean ASA scores, this is why we have scanned medical records for all comorbidities. We have further deleted lines 208-209 and added this to our study flow chart. And added all data from originally in Material and methods to results.
Changes made: page 7-9 lines 277-296, page 8 Figure 4 and page 7 Table 1, page 10 line 332
I recommend improving the discussion section with a more organic exposition, offering a cautious overall interpretation of results, considering objectives, limitations, multiplicity of analyzes and other relevant evidence, placing them in context of previous findings, discussing the generalizability of the study results, and explaining what they mean for future research.
Response: We agree with your suggestion. Our discussion has been unorganized and was exhausting. We have changed the format of the discussion. We discussed our data and considered them in context with our current literature.
Changes made: page 12 line 394-412, line 420-613, line 619-624
Moreover, the authors should add Author Contributions and funding.
Response: We agree with your suggestion.
Changes made: page 14 (line 793-796).
“Author contribution: M.S. and P.M. were the senior surgeons for all patients in this study. J.-P. I. has performed data collection, evaluation and statistical analysis. The finalization of the manuscript was performed by all three authors.”
“No funding was received for this study.”
The paper must be checked by an English native speaker.
Response: We have checked this manuscript by an English native speaker and corrected errors in spelling and grammar.
Reviewer 2 Report
This is a retrospective cohort study aimed to assess if a dedicated stem design might improve clinical and radiographic outcomes of RSA for proximal humeral fractures. The authors conclude that there is not any advantage in using a prosthetic design that theoretically facilitates tuberosity healing.
The study shows relevant limits (retrospective nature, low numerosity of the study groups, high dropout rate, approximate and subjective evaluation of tuberosity healing), that weaken the strength of the research. The authors themselves have highlighted these limits.
However, the interest on the topic and the question raised by the authors counterbalance the weak points of this work.
These are my specific comments:
Patients and methods
- in Table 2 (page 2-3), the authors use a personal classification system to describe fracture morphology. Some points are not clear and probably it would be easier to refer to established systems, such as Neer or AO. For example, they report the term “classical” to describe some fracture patterns, but it’s difficult to understand what is a “classical displaced PHF” or a “classical head split”.
- regarding the description of the surgical technique (page 4):
(L 101-102): was the supraspinatus tendon always resected? Please specify.
(L 103-104): did the authors have any problem in tuberosity reconstruction in presence of a RC tear or fragment comminution? These events are not unusual in the elderly population.
(L 109-110): the authors should specify if the bone grafting technique was the same for both RSA design.
- in classifying tuberosity healing, the authors use the term “extraanatomical healing” (page 5, L 149): it should be changed with “malunion” throughout the manuscript (in Figure 3 caption, in Table 4, at page 9 - L 272)
Results
- (page 8, L 228-229) change “..or an ERLS with greater passive ER greater then active by minimum of..” with “..or an ERLS with passive ER greater then active ER by minimum of..”
- (page 9, L 255) after “integration” add “in RSA”
- (page 9, L 278-280) It’s difficult to explain why LT healing strongly correlates with greater ER. There is theoretical evidence that not repairing the subscapularis may enhance ER, while repairing it may improve internal rotation. The authors should highlight the difficulty in interpreting this observation. The sentence “Possible explanation is better stabilization of the cuff around the prosthesis when tuberosity healing is accomplished” should be removed, also because the mentioned reference (n. 38 - Garofalo et al.) refers to a clinical study in which LT healing was not evaluated.
Conclusion
- (page 10, L 328) The authors state that "ERLS might be an early prognostic factor for clinical outcome after RSA". I would suggest the authors to remove this sentence, unless they can correlate the results of this clinical test with the radiographic appearance of GT, which is probably the ultimate factor influencing the outcome of RSA (thanks to preservation of active ER) and also the main issue investigated in this work.
Author Response
Comments and Suggestions for Authors
This is a retrospective cohort study aimed to assess if a dedicated stem design might improve clinical and radiographic outcomes of RSA for proximal humeral fractures. The authors conclude that there is not any advantage in using a prosthetic design that theoretically facilitates tuberosity healing.
The study shows relevant limits (retrospective nature, low numerosity of the study groups, high dropout rate, approximate and subjective evaluation of tuberosity healing), that weaken the strength of the research. The authors themselves have highlighted these limits.
However, the interest on the topic and the question raised by the authors counterbalance the weak points of this work.
Response:
We thank the reviewer very much for his criticism and appreciating the merits of our work. We further are thankful for the detailed comments and inputs provided to improve our manuscript. We have reviewed each comment and request carefully and have considered each suggestion. We hope that with our revision all questions have been answered and no ambiguities remain. Please find below our answers to the comments and attached also the revised manuscript.
Patients and methods
- in Table 2 (page 2-3), the authors use a personal classification system to describe fracture morphology. Some points are not clear and probably it would be easier to refer to established systems, such as Neer or AO. For example, they report the term “classical” to describe some fracture patterns, but it’s difficult to understand what is a “classical displaced PHF” or a “classical head split”.
Response: We are aware that our personal classification system was confusing. Therefore we have used the classification system by Neer and added it to the first table. Moreover, we have divided the morphologies into the two different groups and emphasized additional head-split fractures or glenoid fractures.
Changes made: all on page 7 Table 1
- regarding the description of the surgical technique (page 4):
(L 101-102): was the supraspinatus tendon always resected? Please specify.
Response: Whenever the SSP was still intact, we have resected its tendon. In 9 cases the SSP was torn or partially torn at point of surgery.
Changes made: page 3, line 142-144 “The supraspinatus was torn or partially torn in nine patients and excised in all others. There were no full-thickness tears of the anterior or posterior rotator cuff present at surgery.”
(L 103-104): did the authors have any problem in tuberosity reconstruction in presence of a RC tear or fragment comminution? These events are not unusual in the elderly population.
Response: Apart from those 9 SSP tears, there were no full-thickness tears of the anterior or posterior cuff present at surgery. Due to our tuberosity refixation technique, where we have placed the sutures at the bone-tendon junction of the tuberosities, the surgeons were able to achieve tuberosity healing even though fragment and even tuberosity comminution were present. We agree with your criticism and have therefore added this information.
Changes made: (page 3, line 141-142) “the osseous-tendon junction to hold …… This allowed an anatomical reduction of tuberosity, even in case of comminution.....
(L 109-110): the authors should specify if the bone grafting technique was the same for both RSA design.
Response: We thank you for your input with regards to the bone grafting technique.
For both stems, we have used cancellous bone from the humeral head underneath the tuberosities to improve chances of healing when thin or egg-shell tuberosities were present. In addition, all fracture specific stems received bone grafting in the metaphyseal window.
Changes made: We have added this on page 4 line 151-53: “Especially, when thin, egg-shell like or comminuted tuberosities were present, cancellous bone was placed underneath. For the fracture-specific stem additional bone grafting within the metaphyseal fenestration was performed.“
- in classifying tuberosity healing, the authors use the term “extraanatomical healing” (page 5, L 149): it should be changed with “malunion” throughout the manuscript (in Figure 3 caption, in Table 4, at page 9 - L 272)
Response: We agree with your suggestion and we have changed the term “extraanatomical healing” to “malunion” throughout the manuscript.
Results
- (page 8, L 228-229) change “..or an ERLS with greater passive ER greater then active by minimum of..” with “..or an ERLS with passive ER greater then active ER by minimum of..”
Response: We agree with this correction.
Changes made: (Page 11, line 372-373) “or an ERLS with passive ER greater than active ER by a minimum..”
- (page 9, L 255) after “integration” add “in RSA”
Response: We agree with this correction.
Changes made: (page 12, line 396): “Although tuberosity integration in RSA is not…”
- (page 9, L 278-280) It’s difficult to explain why LT healing strongly correlates with greater ER. There is theoretical evidence that not repairing the subscapularis may enhance ER, while repairing it may improve internal rotation. The authors should highlight the difficulty in interpreting this observation. The sentence “Possible explanation is better stabilization of the cuff around the prosthesis when tuberosity healing is accomplished” should be removed, also because the mentioned reference (n. 38 - Garofalo et al.) refers to a clinical study in which LT healing was not evaluated.
Response: Thank you very much for your input. Although LT healing did most significantly correlate with ER, LT healing itself cannot biomechanically cause greater ER. As most patient exhibit both LT healing combined together with GT healing, we must come to the conclusion, that this correlation is not a causation.
Changes made: We have removed this sentence and corrected this statement: “A possible explanation is that LT healing was often achieved in combination with GT healing and therefore despite there being a stronger correlation it is not necessarily due to causation that LT healing improves ER.” (page 12, line 428-493).
Conclusion
- (page 10, L 328) The authors state that "ERLS might be an early prognostic factor for clinical outcome after RSA". I would suggest the authors to remove this sentence, unless they can correlate the results of this clinical test with the radiographic appearance of GT, which is probably the ultimate factor influencing the outcome of RSA (thanks to preservation of active ER) and also the main issue investigated in this work.
Response: Thank you very much for your criticism. As we could not correlate the radiographic appearance of GT with the presence of an ERLS successfully in our study population, we have removed this sentence.
Changes made: Sentence is removed (page 13 line 652)